# Comparative Analysis of Frailty Risk and Associated Factors: Community-Based vs. Open Recruitment Groups

**DOI:** 10.3390/ijerph21050611

**Published:** 2024-05-10

**Authors:** Tatsunori Shimizu, Ayuto Kodama, Yu Kume, Masahiro Iwakura, Katsuya Iijima, Hidetaka Ota

**Affiliations:** 1Advanced Research Center for Geriatric and Gerontology, Akita University, Akita 010-8543, Japan; tatsunori@med.akita-u.ac.jp (T.S.); ay-kodama@med.akita-u.ac.jp (A.K.); 2Department of Occupational Therapy, Graduate School of Medicine, Akita University, Akita 010-8543, Japan; kume.yuu@hs.akita-u.ac.jp; 3Department of Environmental Health Science and Public Health, Akita University, Akita 010-8543, Japan; masa.iwaku.91@gmail.com; 4Institute of Gerontology, The University of Tokyo, Tokyo 113-8656, Japan; k-iijima@g.ecc.u-tokyo.ac.jp; 5Institute for Future Initiatives, The University of Tokyo, Tokyo 113-8656, Japan

**Keywords:** frailty risk, oral function, motor function, social function, community-based recruitment

## Abstract

Background: Frailty leads to vulnerability to stress, impaired daily functioning, and an increased need for care. Frailty is considered reversible, and it is crucial to detect the risk of frailty early and investigate factors that may delay its progression. Objectives: To identify tests that can explain frailty risk and compare the situation of local residents with and without frailty support. Methods: Participants were recruited in two ways: through public advertisements in Akita City (open recruitment group) and through invites from frailty supporters in their immediate communities (community-based group). We examined the differences in frailty risk and oral, motor, and social functions between the two groups and identified factors associated with frailty risk in both groups. Results: The community-based group exhibited a lower risk of frailty than the open recruitment group despite having more older members on average. Additionally, the community-based group demonstrated better social functioning than the open-recruitment group. Furthermore, factors such as oral diadochokinesis (ODK), one-leg stand test (OLS), and grip strength (GS) showed significant association with frailty risk. Conclusion: The ODK, OLS, and GS were identified as factors explaining frailty risk, and Frailty Supporters may reduce the risk of frailty.

## 1. Introduction

Aging will progress not only in developed regions but also in developing regions over the next half-century, and aging is a global problem. Japan has shown rapid aging of its population, with the population aged 65 years and over accounting for less than 5% of the total population in 1950 and 28.9% in 2021 [1].

Frailty is a dynamic state affecting an individual who experiences loss in one or more domains of human functioning (physical, psychological, and social). It is caused by the influence of a range of variables and increases the risk of adverse outcomes [2].

Frailty involves not only physical, but also psychological and social aspects [3,4]. Physical frailty includes conditions that make it difficult to perform activities of daily living, such as difficulty in walking due to muscle weakness and deteriorated balance. Psychological frailty includes impaired judgment and cognitive abilities in daily life due to memory impairment and cognitive decline as well as depressed mood and loss of energy and interest due to excessive stress and depression. Social frailty includes increased loneliness and anxiety due to a lack of social connections and support due to isolation; decreased social interaction with family and friends due to aging and limited geographic mobility, making social participation difficult; and a lack of financial stability due to financial problems and housing instability, resulting in a low quality of life. These dimensions are reportedly interconnected [5,6]. In many cases, older individuals pass through intermediate frailty stages and gradually fall into a state of need for nursing care.

The revised Japanese version of the Cardiovascular Health Study (J-CHS) and the Eleven-Check questionnaire are frailty screening tools used in Japan. The revised J-CHS is a frailty screening tool focusing on physical functions (e.g., grip strength and walking speed) [7], whereas the Eleven-Check questionnaire focuses on daily life factors such as nutritional intake and exercise habits. It is not only an effective tool for screening for frailty but also a tool that can help older adults; it plays a role in helping to prevent frailty by improving self-awareness [8]. The two tools approach frailty from different perspectives and play complementary roles.

It is estimated that the proportion of frail and pre-frail individuals in the overall older population in Japan is 7.4–8.7% [9,10] and 40.8–48.1% [9,10], respectively, and that approximately 50% of the older population is either frail or pre-frail.

When considering the relationship between aging and frailty, it is essential to move beyond merely viewing aging as a natural process and to understand its broader impact. Age-related decline in physical function and metabolic capacity may increase the risk of developing frailty. Frailty not only leads to a decrease in an individual’s quality of life, but also contributes to an increase in the societal economic burden due to the rising need for caregiving and support. Therefore, in the context of advancing aging, preventing and intervening early frailty has become a critical challenge. The risk factors for the onset and progression of frailty encompass a wide range of factors, including demographic, social, clinical, lifestyle, and biological factors [11]. However, frailty is reversible, and a healthy state can be restored with appropriate intervention [12]. Thus, for modifiable factors, the early detection and implementation of proactive preventive measures are expected to facilitate the prevention and improvement of frailty.

Interest in early detection of frailty has increased in recent years. The number of research papers using various sensors and devices (e.g., smartphones) to gather information to help diagnose frailty was ten times greater in 2020 than in 2010 [13]. In addition, recent reports have shown a relationship between oral function and physical frailty, with oral function decline affecting physical frailty [14] and oral health variables correlating with frailty scale scores [15]. In a previous study, we demonstrated the correlation between decreased oral function and frailty [16]. We report that the evaluation of oral function is crucial for the early prevention of frailty. Testing methods that are not limited to oral function, are relatively easy to perform without specialists, and can assess frailty risk, would greatly help in the early detection of frailty.

We are aware of the importance of social networks and community awareness-raising activities in the prevention of frailty. In Japan, people who received training on frailty, learned proactively and are capable of conducting examinations for frailty and providing health guidance. These are trained by the general public and are called Frailty Supporters. The participants were citizen volunteers who conducted and administered frailty awareness activities and tests in their communities. In Akita City, 57 people worked as active Frailty Supporters as of April 2024, and this number is increasing. Globally, professionals are often responsible for frailty prevention and early detection activities. Few studies have examined the effects of community volunteers who learn about frailty in their communities.

The purpose of this study was two-fold: one was to identify items associated with frailty risk by testing/questioning all participants for oral, motor, and social functions, which is easy to perform. The second was to compare the situation of local residents with and without Frailty Supporters. Specifically, the study population was divided into two groups: a group of participants recruited from the public in Akita City (open recruitment group) and residents of the community where the Frailty Supporters live (community-based group), and the aforementioned functional tests and frailty risks were be examined. The social ties in the community created around Frailty Supporters and effective examinations that can be conducted autonomously in the community were considered ideal for the prevention and early detection of frailty.

## 2. Materials and Methods

### 2.1. Participants

The participants were recruited from Akita City, Akita Prefecture, Japan. Participants were recruited in two ways: first, through public advertisements in Akita City (open recruitment group) and second, through invites from frailty supporters in their immediate communities (community-based group). In the open recruitment group, 402 individuals participated, with 397 included in the analysis, with an average age of 73.9 ± 7.9 years (age range 25–92). In the community-based group, 420 individuals participated, with 417 included in the analysis, with an average age of 75.4 ± 7.0 years (age range 29–93). The sample size was calculated using G*Power [17]. For samples in the present study, the estimated sample size was 676 participants to detect a clinically significant effect with the number of groups = 2, α = 0.05, power = 90% and effect size = 0.25 for difference between two independent means (two groups) [17]. The inclusion criteria for this study were as follows, participant age: 20 years or older; residence: Akita City. The exclusion criteria were difficulty in reading and writing, and presence of quadriplegia due to illnesses such as stroke. The study was conducted from 19 November 2021, to 12 January 2022. The study was conducted with the approval of the ethics committee (approval no. 1649). In this study, occupational therapists and doctors were involved in the research and served as backups for Frailty Supporters.

### 2.2. Assessment of Risk of Frailty

Patients with frailty were evaluated using the Eleven Check questionnaire developed by the Institute of Gerontology at the University of Tokyo [18,19] (Appendix A). The questionnaire comprises 11 distinct elements: two pertaining to dietary habits, two concerning oral functionality, three addressing motor skills, and four focusing on social and cognitive functions, all pivotal to maintaining optimal health. Responses were dichotomous, categorized as either “yes” or “no”, with a score of 1 assigned for favorable habits and 0 for unfavorable habits, resulting in a total score ranging from 0 to 11 (where high scores denoted low frailty risk). Additionally, individuals scoring 6 or above were classified into the “No risk of frailty” category, while those scoring between 0 and 5 were categorized into “high risk of frailty” category.

### 2.3. Assessment of Oral Function

Oral function was assessed using two measures: oral diadochokinesis (ODK) and quality of life (QOL). ODK was gauged using the “Kenkou-kun Handy” automatic measuring device (Takei Scientific Instruments Co., Ltd., Niigata, Japan), which recorded the number of repetitions of the monosyllables “ta” or “ka” per second. Quality of life associated with oral function was evaluated using the General Oral Health Index (GOHAI) [20], a self-administered questionnaire. Each question featured five response categories with corresponding scores (1 = always, 2 = often, 3 = sometimes, 4 = rarely, and 5 = never) assigned to each category. To maintain consistency, scores from the positively worded questions were reversed during data analysis. The GOHAI score represents the sum of the responses to the 12 questions, with a high score (maximum = 60) indicating “good oral health,” reflecting a favorable state of internal health.

### 2.4. Assessment of Motor Function

Motor function was assessed using multiple parameters, including the one-leg stand test (OLS), lower leg circumference (LLC), grip strength (GS), and Skeletal Muscle Mass Index (SMI). During the OLS test, the participants sat on a chair, lifted their nondominant leg off the ground, and stood up without assistance. The examiners determined whether the subjects could maintain a one-leg standing posture for at least 3 s without support. The LLC was measured by placing a measuring tape around the thickest part of the non-dominant lower leg while the participants were seated. The GS was quantified using a Smedley-type handheld dynamometer (Takei Corporation, Niigata, Japan). The SMI was computed by assessing the skeletal muscle mass in the limbs using a body composition analyzer (Inner Scan Dual RD-800, TANITA Corporation, Tokyo, Japan) and dividing it by the square of the height (m).

### 2.5. Assessment of Social Functions

Participants’ social functioning was evaluated with regard to their interpersonal connections, organizational engagement, and social support. Interpersonal connections were assessed using a self-administered questionnaire based on the Japanese version of the Lubben Social Network Scale shortened version (LSNS-6) [21]. This scale, which is tailored for older adults, comprises three items each related to family networks and three related to non-family networks. Total scores, ranging from 0 to 30, represent an equally weighted sum of scores for the six items, with high scores indicating strong social networks and scores below 12 suggesting social isolation. Organizational participation was determined by counting the number of affirmative responses to seven items, employing a binary ‘yes’ or ‘no’ method to indicate membership in various organizations, such as senior citizens’ clubs or circles: (i) senior citizens’ associations; (ii) health/sports organizations (other than senior citizens’ associations); (iii) learning/education organizations (other than senior citizens’ associations); (iv) hobby organizations (other than senior citizens’ associations); (v) neighborhood councils; (vi) volunteer organizations; and (vii) income-generating companies. Social support was assessed using a two-part ‘yes’ or ‘no’ method to determine both the support received by the participant and the support extended by the participant to others. The total score was calculated based on the number of affirmative responses to four items: (i) availability of someone to listen to concerns and complaints; (ii) availability of someone to assist with household tasks, shopping, and care/nursing; (iii) willingness to listen to others’ concerns or complaints; and (iv) willingness to assist others with household tasks, shopping, and care/nursing.

### 2.6. Statistical Analysis

Parametric tests were used to compare age, oral function (ODK and GOHAI), motor function (LLC, GS, and SMI), and social function (LSNS-6, organizational participation, and social support) between the open recruitment and community-based groups. Sex, occlusal force, OLS, and scores on the Eleven Check questionnaire were compared using the chi-square test. A binomial logistic regression analysis was conducted to determine the factors associated with frailty risk as the dependent variable. The independent variables for the regression model included age, sex, occlusal force, ODK, GOHAI, OLS, LLC, GS, LSNS-6, organizational participation, and social support. The analysis was performed using SPSS Version 27.0 for Windows (SPSS Inc., Chicago, IL, USA), with a significance level set at *p* = 0.05.

## 3. Results

### 3.1. The Community-Based Group Showed a Lower Risk of Frailty than the Open Recruitment Group

First, we compared age, sex, and results of the Eleven Check questionnaire between the open recruitment and community-based groups. As shown in Table 1, the community-based group had a significantly older average age than the open recruitment group. While both groups had a high proportion of women, no significant differences were observed between them. In this study, the level of frailty risk was assessed using the total score of the Eleven Check questionnaire. The total scores were higher in the open recruitment group than in the community-based group. These results indicate that the risk of frailty was lower in the community-based group than in the open recruitment group, despite older age.

### 3.2. The Social Functions of the Community-Based Group Had Better Scores than Those of the Open Recruitment Group

Table 2 shows the oral, motor, and social functions in both groups. We observed that ODK had a significantly better performance (6.7 < 6.5 times/s) in the open recruitment group compared to the community-based group, while no differences were observed in other items of oral functions. There were no significant differences in motor function between the two groups. Regarding social functions, the community-based group scored significantly better for connections with others, organizational participation, and social support than the open recruitment group.

### 3.3. ODK, OLS, and GS Had a Significant Association with Physical Frailty

Finally, in order to clarify factors predicting frailty risk in all subjects, we conducted a binomial logistic regression analysis with the presence of physical frailty as the dependent variable and age, sex, occlusal force, ODK, GOHAI, OLS, LLC, GS, LSNS-6, organizational participation, and social support as independent variables. The analysis was performed by pooling data from both open recruitment and community-based groups. Owing to the high correlation coefficients between LLC and SMI as well as between GS and SMI, SMI was excluded as an independent variable. Binomial logistic regression analysis indicated a significant association between the presence of physical frailty and ODK (odds ratio, 1.289; 95% confidence interval (95% CI), 1.054–1.577; *p* = 0.014), OLS (odds ratio, 2.119; 95% CI, 1.520–2.954, *p* < 0.001), and GS (odds ratio, 1.031; 95% CI, 1.003–1.060; *p* = 0.03) (Table 3). Furthermore, the regression model demonstrated a good fit, as evidenced by the results of the Hosmer–Lemeshow test (*p* = 0.923), with a high percentage of correct classifications at 74.8%.

## 4. Discussion

In this study, we compared the risk factors for frailty (oral, motor, and social functions) between open recruitment and community-based groups. In Table 1, we observe that the community-based group, despite having a higher average age, had a lower total score on the Eleven Check questionnaire than did the open recruitment group. Among the people in the community-based groups, communities were formed around Frailty Supporters. These findings indicate that the risk of frailty was lower in the community-based group than in the open recruitment group, despite older age. According to these results, we suggest that the risk factors for frailty may be not only age but also community-based lifestyle and support systems. As previously reported, physical frailty has been shown to encompass multifaceted issues including psychological and social factors. [3,4] Thus, its association with physical activity, cognitive function, and community engagement has been documented [22].

In this study, the community-based group scored significantly better for connections with others, organizational participation, and social support than the open recruitment group (Table 2). The better social functions observed in the community-based group are consistent with previous reports, emphasizing the importance of social engagement in mitigating frailty risks [23]. This finding highlights the potential benefits of community-based interventions in promoting social connectedness and overall well-being. Future research should include a detailed evaluation of how frailty supporters improve social functioning. This included an investigation of how local support networks and community events promote social functioning.

Several studies have reported that social frailty is associated with physical frailty [24,25] and oral frailty [26,27]. In our study, no significant differences in motor function were found between the two groups (Table 2). Moreover, contrary to our expectations, ODK had a significantly better performance (6.7 < 6.5 times/s) in the open recruitment group (Table 2). Future research is needed to understand why this result occurs in a community-based recruitment, which is considered to have a lower overall risk of frailty and better social functioning. Specifically, the lack of significant differences in motor function requires a detailed analysis of motor function in future studies. This analysis includes comparisons of specific motor function subscales, the impact of motor function on daily living, and the interaction of motor function with other frailty risk factors. To elucidate the unexpected result of better ODK performance in the open recruitment group, it is important for future research to investigate the factors that contribute to ODK. These include factors such as oral and lingual muscle strength and mobility, voice control function, and oral environment.

Finally, a binomial logistic regression analysis identified the ODK, OLS, and GS as significant factors associated with physical frailty, highlighting their importance in assessing and addressing frailty in older populations. In this study, although age showed significant correlations with ODK, OLS, and GS (−0.31 (*p* < 0.01), 0.328 (*p* < 0.01), −0.199 (*p* < 0.01)), the correlation coefficients are not high. Binomial logistic regression analysis showed that age was not associated with the risk of frailty. Therefore, we suggest that frailty risk could be predicted by the ODK, OLS, and GS, and might not be solely mediated by age.

To summarize, one of our objectives was to conduct relatively easy-to-perform tests/questions regarding oral, motor, and social functions and frailty risk, and to identify items associated with frailty risk. The second was to compare the situation of community residents with and without Frailty Supporters. For the former, the ODK, OLS, and GS were extracted as factors that could explain frailty risk. For the latter, people in the community-based group (i.e., those with frailty support) were significantly older than those in the open recruitment group (i.e., those without frailty support); however, their risk of frailty was significantly lower. Although the open-recruitment group had better results in terms of oral function, the community-based group had better social function. These results observed in the community-based group may be due to the improvement in the social function of the local residents through the activities of Frailty Supporters. The activities of Frailty Supporters, who are citizen volunteers, are thought to reduce the risk of frailty not only by educating local residents, but also by improving the social activities of residents through educational opportunities. Based on the results of this study, public health policies should include strengthening community-based lifestyles and support systems. Additionally, public health policies should promote the use of simple assessment tools in communities that can be used without the need for specialists. Frailty-screening activities conducted by health professionals and community volunteers can support early detection and intervention within communities.

This study has several limitations. First, the participants were predominantly female, which may have affected our results. Second, participants in the open recruitment group were self-selected to participate, whereas participants in the community-based group were invited by neighbors, who may have selected relatively healthy individuals. Third, because of the nature of the community-based group, which invited neighbors, and because the testing was conducted within walking distance of each community, a large percentage of participants were over 80 years of age (24.2% and 29.3%, respectively). In addition, although the cause was unclear, a large percentage of participants in the open recruitment group were under 60 years of age (3.5% and 1.2%, respectively). These factors may have influenced the results. Finally, we were unable to interview participants about factors that might influence frailty, such as marital status, family members living in the same household, ties to third parties, place of residence, and education.

## 5. Conclusions

In this study, ODK, OLS, and GS were identified as factors that explained frailty risk. The community-based group had a significantly lower frailty risk and significantly better social functioning than the open recruitment group. This suggests that the activities of Frailty Supporters, who are citizen volunteers, may reduce the risk of frailty not only by educating local residents but also by improving their social activities.

## Figures and Tables

**Table 1 ijerph-21-00611-t001:** Comparison of age, sex and Eleven Check in the open recruitment group and community-based group.

	Open Recruitment			Community-Based			*p*-Value
	Mean		SD	Mean		SD	
Age (year)	73.9		7.9	75.4		7.0	0.005 *
Sex (% female)		77.3			78.2		0.771
Eleven Check (% risk)		29			22.8		0.044 *

* *p* < 0.05, the parametric test (age), the *X*^2^ test (gender, Eleven Check).

**Table 2 ijerph-21-00611-t002:** Comparison of oral, motor, and social functions in the open recruitment group and community-based group.

	Open Recruitment			Community-Based			*p*-Value
	Mean		SD	Mean		SD	
Oral Functions							
Occlusal force (%)		95.7			93.3		0.129
Oral diadochokinesis (times/s)	6.7	6	0.8	6.5		0.8	<0.001 **
General Oral Health Index (point)	53.8		6.5	54.5		5.9	0.127
Motor Functions							
One-leg stand test (%)		57.2			62.6		0.115
Lower leg circumference (cm)	34.5		2.8	34.5		3.0	0.728
Grip strength (kg)	24.7		6.2	24.2		6.5	0.273
Skeletal Muscle Mass Index	6.73		0.7	6.7		0.8	0.692
Social Functions							
Connection with others (score)	17.0		5.8	18.4		5.0	<0.001 **
Organizational participation (score)	3.0		1.8	3.3		1.6	0.006 *
Social support (score)	3.5		0.9	3.6		0.7	0.040 *

* *p* < 0.05, ** *p* < 0.01, the parametric test (oral diadochokinesis, General Oral Health Index, lower leg circumference, grip strength, Skeletal Muscle Mass Index, connection with others, organizational participation, and social support), the *X*^2^ test (occlusal force, one-leg stand test).

**Table 3 ijerph-21-00611-t003:** Models according to binomial logistic regression analysis.

	Coefficient (B)	Odds Ratio	95% CI	*p*-Value
Oral diadochokinesis	0.254	1.289	1.054, 1.577	0.014 *
One-leg stand test	0.751	2.119	1.520, 2.954	<0.001 **
Grip Strength	0.031	1.031	1.003, 1.060	0.03 *

* *p* < 0.05, ** *p* < 0.01, Hosmer-Lemeshow test *p* = 0.923, percentage of correct classifications 74.8%.

## Data Availability

Data are available on request owing to restrictions such as privacy or ethics. Data presented in this study are available upon request from the corresponding author.

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
