# Peer review of "Comparative Analysis of Frailty Risk and Associated Factors: Community-Based vs. Open Recruitment Groups"

_ijerph, 2024, doi:10.3390/ijerph21050611_

Round 1

Reviewer 1 Report

Comments and Suggestions for Authors

very interesting research, I have some questiones:

explain how did you calculated your sample size (very important)

introduction is good but you can be little extensive

in tables sohw means±sd instead in separated columns

could you show the ROC graph and AUC for binomial logistic model?

Author Response

RESPONSE TO REVIEWER 1:

Thank you for your review of our paper. We have answered each of your points below.

Comment 1:

explain how did you calculated your sample size (very important)

Response:

We thank the reviewer for this comment. Sample size was calculated using G*Power[Faul et al]. we have added the following text to the Materials and Methods (Line 112-116).

Comment 2:

introduction is good but you can be little extensive

Response:

We thank the reviewer for this comment. The introduction has been revised and added, including some points that were not explained well enough.

Comment 3:

in tables sohw means±sd instead in separated columns

Response:

Thank you very much for pointing this out. I was concerned that the results of the χ-square test would be difficult to understand if they were listed in ±SD in the same table. I am very sorry, but may I leave it as it is?

Comment 4:

could you show the ROC graph and AUC for binomial logistic model?

Response:

The ROC curve and AUC for the binomial logistic model are shown below.

AUC=0.647

Reviewer 2 Report

Comments and Suggestions for Authors

Review:

Thank you for the opportunity to review your article. Below, I provide some contributions for its improvement.

The objective is not explicitly stated in the summary section, it should be incorporated.

In the introduction, I suggest a review and overview of the structure to make it more organized and improve readability and understanding. Improvements in terminology, such as "Frailty is a multidimensional concept that refers not only to physical but also psychological and social domains," could be expressed more directly, since "Fragility involves not only physical aspects but also psychological and social. I would have liked to see the incidence and prevalence of frail patients not only in the literature but also in the introduction, characteristics of the patients, etc.

The methodology used to assess the risk of frailty and associated factors in older adults in Akita, Japan, appears appropriate, but could be improved by providing details on the inclusion/exclusion criteria and demographic characteristics of the sample, as well as more information on the recruitment methods.

The discussion could benefit from clearer interpretations and more detailed explanation of unexpected results. I suggest expanding discussions on the limitations of the studies and suggesting specific areas for future research.

It would be beneficial to highlight more clearly how the study findings have practical implications for healthcare professionals and how they may influence future interventions to prevent frailty, which healthcare professionals would be responsible for this practice. Additionally, specific recommendations based on the study results could be proposed to guide public health policies, as well as community intervention strategies led by health professionals.

In this study, it is not clear which health professionals are conducting the research. It would be interesting to mention them to better justify the need for the study.

Good job from a public health perspective, thank you.

Author Response

RESPONSE TO REVIEWER 2:

Thank you for your review of our paper. We have answered each of your points below.

Comment 1:

The objective is not explicitly stated in the summary section, it should be incorporated.

Response:

Thank you for pointing this out. I added the objective in the abstract (Line 15,16).

Comment 2:

In the introduction, I suggest a review and overview of the structure to make it more organized and improve readability and understanding. Improvements in terminology, such as "Frailty is a multidimensional concept that refers not only to physical but also psychological and social domains," could be expressed more directly, since "Frailty involves not only physical aspects but also psychological and social. I would have liked to see the incidence and prevalence of frail patients not only in the literature but also in the introduction, characteristics of the patients, etc.

Response:

We thank the reviewer for this comment. We have made significant revisions to the introduction to improve readability and comprehensibility (Line 34-101). We have changed the use of the terminology you suggested (Line 37) and described the incidence of frailty (Line 57-59) and the characteristics of patients (Line 38-46).

Comment 3:

The methodology used to assess the risk of frailty and associated factors in older adults in Akita, Japan, appears appropriate, but could be improved by providing details on the inclusion/exclusion criteria and demographic characteristics of the sample, as well as more information on the recruitment methods.

Response:

Thank you for pointing out this important point. Inclusion/exclusion criteria (Line 116-118) and recruitment methods (Line 106-109) have been described. Demographic characteristics such as education level, marital status and family members living in the same household were not collected in this study and this is noted in the limitation (Line 305-307).

Comment 4:

The discussion could benefit from clearer interpretations and more detailed explanation of unexpected results. I suggest expanding discussions on the limitations of the studies and suggesting specific areas for future research.

Response:

I apologize for the inadequate discussion. We have tried to be clear in our interpretations, and have added discussion of unexpected results (Line 299-305) and specific suggestions for future research (Line 261-268).

Comment 5:

It would be beneficial to highlight more clearly how the study findings have practical implications for healthcare professionals and how they may influence future interventions to prevent frailty, which healthcare professionals would be responsible for this practice. Additionally, specific recommendations based on the study results could be proposed to guide public health policies, as well as community intervention strategies led by health professionals.

Response:

Thank you very much for your very important remarks.

We apologize for the lack of explanation at the time of our initial submission. The community-based group in this study is the residents of the community where the Frail Supporters are located. The Frailty Supporters are citizen volunteers who have taken a course on frailty, learned about frailty themselves, and are in charge of frailty awareness and frailty testing. This means that this study will show the effect of Frailty Supporters on the prevention of frailty for local residents and which factors of local residents are influenced by Frailty Supporters.

Specific recommendations based on research results are added to the discussion (Line 290-295).

Comment 6:

In this study, it is not clear which health professionals are conducting the research. It would be interesting to mention them to better justify the need for the study.

Response:

In this study, occupational therapists and doctors are involved in the research. This has been added to Materials and Methods (Line 120-122).

Reviewer 3 Report

Comments and Suggestions for Authors

- The introduction is short and unclear. Firstly, the definition of frailty is questionable. I recommend reading the works of Gobbens et al. (2010, 2021). Then, the assertion that aging is a problem. Aging is universal, and it is not clear why it would be considered a problem. The gaps in the literature that the authors intend to fill are not clear, as there is no discussion of previous studies that adds the current one to the literature. The only reference is from a study by the authors, a self-citation (reference 8), which raises doubts about the current research. The authors do not clearly present the need for the study, as the type of recruitment is not a valid justification (see next comment). The absence of hypotheses also raises questions about the need for the study, as the authors do not even discuss possible differences in frailty based on the type of recruitment.

- The authors should better characterize the sample by presenting the sample size calculation, age, age range, and standard deviation, inclusion and exclusion criteria, other important sociodemographic characteristics such as gender, place of residence, marital status, educational qualifications, as these variables influence frailty.

- The subchapter on data collection procedures and how the data were collected is missing. In fact, the variable that seems to justify the study is the type of participant recruitment, which suggests, due to their similarity, that any participant could be placed in either group. That is, it is not clear what distinguishes the groups, as a participant could fit into either one. Additionally, due to the test battery, there is a lack of explanation of who collected the data, experience in physical tests, the data collection sequence based on sample size, the execution criteria of the tests based on age (given the average age, did all participants have the literacy skills to complete written tests?). These are all questions that need to be clarified in the document.

- The authors highlight several collected variables, but these are not discussed in the introduction, which reinforces the lack of understanding of the current study. Additionally, more information is needed on test administration. For example, in grip strength, how many attempts were made? In what body position?

- Why was the Mann-Whitney test used? Considering the sample size, parametric tests seem favorable. Since the objective is not clear, the logistic analysis is not understood.

- The correct term is sex, not gender, if authors refer to biological issues.

- I question how the authors compared the sample size of the % of female sex based on the type of recruitment, which seems statistically impossible. As well as the use of the frailty indicator as dichotomous. Statistically, it is wrong.

- The same in Table 2 when authors use dichotomous variables as both independent and dependent variables simultaneously. Also in this table, the results are expected, as the comparison variable is recruitment. In fact, the only differences exist in the social aspect, however, this may be influenced by marital status, family or third-party connections, something that is hardly influenced by the method of participant recruitment for the study.

- The discussion is biased based on the results, which, as mentioned earlier, are incorrect because the authors use qualitative data as independent and dependent. Even with the chi-square test, this only indicates possible differences but does not indicate the direction or intensity of the differences.

- The entire discussion needs to be revised based on everything mentioned above. However, considering the current data and the study that was conducted, I have some doubts about the feasibility of the study.

- The conclusions are completely out of context. The authors cannot claim ", we have confirmed that a community-based recruitment method may be effective in preventing frailty, and that some oral and physical function assessments could predict the risk of Frailty" when the study is cross-sectional and no tests were conducted to prove this statement. This assertion "Encouraging many elderly people to take part in easy tests that predict frailty risk by community-based recruitment, and then providing interventions tailored to the community to address these risks, can help prevent frailty." is speculative, as it was not the aim of your study and you do not have data to support this claim.

Comments on the Quality of English Language

needs revisions.

Author Response

RESPONSE TO REVIEWER 3:

Thank you for your review of our paper. We have answered each of your points below.

Comment 1:

The introduction is short and unclear. Firstly, the definition of frailty is questionable. I recommend reading the works of Gobbens et al. (2010, 2021). Then, the assertion that aging is a problem. Aging is universal, and it is not clear why it would be considered a problem. The gaps in the literature that the authors intend to fill are not clear, as there is no discussion of previous studies that adds the current one to the literature. The only reference is from a study by the authors, a self-citation (reference 8), which raises doubts about the current research. The authors do not clearly present the need for the study, as the type of recruitment is not a valid justification (see next comment). The absence of hypotheses also raises questions about the need for the study, as the authors do not even discuss possible differences in frailty based on the type of recruitment.

Response:

Thank you very much for your valuable suggestions.

The introduction has been rewritten to add explanations that were insufficient in the initial submission and to make it clearer so that it is not poorly understood. The community-based group is a group whose participants are residents of a community with a Frailty Supporter who invited each other to participate, while the open recruitment group is one in which people with no such connections participated. The different recruitment methods in this study differed in the presence or absence of involvement with Frail Supporters, and we would have expected to see an effect of frail suppression in the community group.

Aging comes to everyone, and this in itself is not a problem, but the inevitable physical and metabolic decline associated with aging is a risk factor for frailty (Line 60-63), and we recognize that prevention and early detection of frailty for this inevitable risk factor is our challenge. References to recent studies (reference 13-15) and the need for this study are added in the introduction (Line 89-91).

Comment 2:

The authors should better characterize the sample by presenting the sample size calculation, age, age range, and standard deviation, inclusion and exclusion criteria, other important sociodemographic characteristics such as gender, place of residence, marital status, educational qualifications, as these variables influence frailty.

Response:

Thank you for pointing out this important point. The sample size calculation (Line 112-116), age range (Line 110-111, Line 112), standard deviation (Line 110,112), and inclusion and exclusion criteria (Line 116-118) have been added. Other factors such as place of residence, marital status, and education were not investigated in this study. These are listed in the limitation section, along with the fact that they are factors that may influence frailty (Line 305-307).

Comment 3:

The subchapter on data collection procedures and how the data were collected is missing. In fact, the variable that seems to justify the study is the type of participant recruitment, which suggests, due to their similarity, that any participant could be placed in either group. That is, it is not clear what distinguishes the groups, as a participant could fit into either one. Additionally, due to the test battery, there is a lack of explanation of who collected the data, experience in physical tests, the data collection sequence based on sample size, the execution criteria of the tests based on age (given the average age, did all participants have the literacy skills to complete written tests?). These are all questions that need to be clarified in the document.

Response:

Thank you very much for your very important remarks.

We apologize for the lack of explanation at the time of our initial submission. The community-based group in this study is the residents of the community where the Frail Supporter is located, and the community residents were invited to participate in this study by the Frailty Supporter. (The Frailty Supporters are citizen volunteers who have taken a course on frailty, learned about frailty themselves, and are in charge of frailty awareness and frailty testing.) Therefore, it is not likely that participants will be in either group. Information how to participate (Line 106-109) has been added. The inspection procedure is described in Materials and Methods (Line 147-157). As for whether all participants had the literacy skills to complete the written test, only those who could read and write participated were in this study (Line 117-118).

Comment 4:

The authors highlight several collected variables, but these are not discussed in the introduction, which reinforces the lack of understanding of the current study. Additionally, more information is needed on test administration. For example, in grip strength, how many attempts were made? In what body position?

Response: 

Thank you very much for pointing out these important points. The characteristics of the criteria for frailty in Japan and the characteristics of the frailty test conducted in this study were added in the introduction. In addition, a recent paper on the validity of the frailty test used in this study is cited (Line 49-56).

Comment 5:

Why was the Mann-Whitney test used? Considering the sample size, parametric tests seem favorable. Since the objective is not clear, the logistic analysis is not understood.

Response:

Thank you very much for pointing out this important point.

In fact, we are performing a parametric test, not a Mann-Whitney test. We have corrected the description of Materials and Methods and each table.

Comment 6:

The correct term is sex, not gender, if authors refer to biological issues.

Response:

This error has been corrected in accordance with the reviewer's comment (Line 183, 187, 193, 220, Table1).

Comment 7:

I question how the authors compared the sample size of the % of female sex based on the type of recruitment, which seems statistically impossible. As well as the use of the frailty indicator as dichotomous. Statistically, it is wrong.

Response:

Thank you for pointing this out. The table shows %female, but the χ-square test was performed using the number of males and females in each group.

Comment 8:

The same in Table 2 when authors use dichotomous variables as both independent and dependent variables simultaneously. Also in this table, the results are expected, as the comparison variable is recruitment. In fact, the only differences exist in the social aspect, however, this may be influenced by marital status, family or third-party connections, something that is hardly influenced by the method of participant recruitment for the study.

Response:

Thank you very much for pointing out this important point.

I apologize for the lack of explanation in my initial post. This is a comparison of two groups with different recruitment methods, but the nature of the two groups can be thought of as areas with or without the involvement of frail supporters. We think it is very interesting to note that the socialization item was favorable in the comparison of the two groups. This study did not examine marital status, family or third-party ties, and other factors that may influence frailty, so we have included these in the limitations section (Line 304-307).

Comment 9:

The discussion is biased based on the results, which, as mentioned earlier, are incorrect because the authors use qualitative data as independent and dependent. Even with the chi-square test, this only indicates possible differences but does not indicate the direction or intensity of the differences. The entire discussion needs to be revised based on everything mentioned above. However, considering the current data and the study that was conducted, I have some doubts about the feasibility of the study.

Response:

Thank you for pointing this out. In a binomial logistic regression analysis, we believe it is possible to throw in qualitative data as independent variables. When qualitative data are used in a dichotomous logistic regression analysis, dummy variables are usually created and used in the analysis, and we have done the same in this study. This allows the qualitative data to be incorporated into the dichotomous logistic regression model and its impact to be assessed.

Comment 10:

The conclusions are completely out of context. The authors cannot claim ", we have confirmed that a community-based recruitment method may be effective in preventing frailty, and that some oral and physical function assessments could predict the risk of Frailty" when the study is cross-sectional and no tests were conducted to prove this statement. This assertion "Encouraging many elderly people to take part in easy tests that predict frailty risk by community-based recruitment, and then providing interventions tailored to the community to address these risks, can help prevent frailty." is speculative, as it was not the aim of your study and you do not have data to support this claim.

Response:

I apologize for the lack of explanation in my initial post.

The purpose of this study is two-fold: one is to find items associated with frailty risk by testing/questioning all participants for oral, motor, and social function, which are relatively easy to perform, and for frailty risk. The other is to compare the situation of local residents with and without Frailty Supporters. The two groups can be distinguished by differences in recruitment methods. The community-based group is a group whose participants are residents of a community with a Frailty Supporter who invited each other to participate, while the open recruitment group is one in which people with no such connections participated. Frailty Supporters are volunteers in the community who are trained and expected to learn about frailty themselves and to raise awareness about frailty in the community and conduct frailty screenings. The results of this study showed a significantly lower risk of frailty and better social facter in the community group. From this point of view, autonomous activities in the community were considered to be important for the prevention of frailty. We believe that it is important to examine simple tests that can be performed without a specialist for autonomous activities, and we believe that the results of this study will help in this regard.

Round 2

Reviewer 1 Report

Comments and Suggestions for Authors

the paper has been improved, only few questions
reference where you take an effect size of 0.25 to calculate your effect size

Author Response

RESPONSE TO REVIEWER 1:

Thank you for your review of our paper. We have answered each of your points below.

Comment :

the paper has been improved, only few questions

reference where you take an effect size of 0.25 to calculate your effect size

Response:

Thank you very much for your suggestion.

The calculation of the effect size was done with G*Power, and the method of calculating the effect size with G*Power is described in reference 17. To make this clear, we have inserted the reference 17.

Reviewer 3 Report

Comments and Suggestions for Authors

The authors did a good job in reviewing their manuscript. I have no further comments.

Comments on the Quality of English Language

Minor spelling check.

Author Response

RESPONSE TO REVIEWER 3:

Thank you for your review of our paper. We have answered each of your points below.

Comment :

The authors did a good job in reviewing their manuscript. I have no further comments.

Response:

I would like to thank you from the bottom of my heart for your previous suggestions, which helped us to improve our paper to a better one.

Comments on the Quality of English Language: Minor spelling check:

Response:

Thank you for pointing this out. We have corrected some errors.